# Direct assessment of possible mutations in the 23S rRNA gene encoding macrolide resistance in *Chlamydia trachomatis*

J. M. van Niekerk[1], I. H. M. van Loo[1], M. Lucchesi[1,2,3], S. A. Morré[2,3,4,5], C. J. P. A. Hoebe[1,2,3,6,7], N. H. T. M. Dukers-Muijrers[6,8], P. F. G. Wolffs[1,2,3]*

1 Department of Medical Microbiology, Care and Public Health Research Institute (Caphri), Maastricht University Medical Center, Maastricht, The Netherlands, 2 Dutch *Chlamydia trachomatis* Reference Laboratory, Department of Medical Microbiology, Care and Public Health Research Institute (Caphri), Maastricht University Medical Center, Maastricht, Netherlands, 3 Laboratory of Immunogenetics, Department of Medical Microbiology and Infection Control, VU Medical Center, VU University Medical Center, Amsterdam, The Netherlands, 4 Institute of Public Health Genomics, Genetics & Cell Biology Cluster, GROW Research School for Oncology and Developmental Biology, Faculty of Health Medicine and Life Sciences, Maastricht University, Maastricht, The Netherlands, 5 Department of Molecular and Cellular Engineering, Jacob Institute of Biotechnology and Bioengineering, Sam Higginbottom University of Agriculture, Technology and Sciences, Allahabad, Uttar Pradesh, India, 6 Department of Sexual Health, Infectious Diseases and Environmental Health, South Limburg Public Health Service, Geleen, The Netherlands, 7 Department of Social Medicine, Care and Public Health Research Institute (Caphri), Maastricht University, Maastricht, The Netherlands, 8 Department of Health Promotion, Care and Public Health Research Institute (Caphri), Maastricht University, Maastricht, The Netherlands

* p.wolffs@mumc.nl

## Abstract

Reports of potential treatment failure have raised particular concerns regarding the efficacy of the single dose azithromycin regimen in the treatment of urogenital and anorectal *Chlamydia trachomatis* (CT) infections. Several factors have been suggested, including heterotypic resistance. Antimicrobial susceptibility testing in CT requires cell culture with serial dilutions of antibiotics, which is laborious and for which there is no standardized testing methodology. One method to partly overcome these difficulties would be to use a genotypic resistance assay, however most current available assays do still require prior CT culture. In order to facilitate the assessment of genotypic resistance directly from clinical samples, without the need for prior culture, the aim of this study was to develop a CT specific PCR assay for the assessment of resistance associated mutations (RAMs) in the 23S rRNA gene, and to evaluate a sample of clinical cases in which CT PCR's remained positive during follow-up despite azithromycin treatment. Neither the *in silico* analysis nor the analytical specificity testing demonstrated clinically relevant cross-reactivity with other bacterial species. These results in conjunction with the analytical sensitivity demonstrating consistent CT 23S rRNA gene detection in the range of 10e3 IFU/mL, exemplify the assay's apt performance. Although no known macrolide RAMs were detected in the clinical cases, the described assay allows future culture independent macrolide RAM surveillance in CT, and increases accessibility for other laboratories to engage in screening.

**Data Availability Statement:** All 23S sequences obtained in this study are available in NCBI GenBank with the accession numbers OM320821-OM320875. S3 Table includes the corresponding accession numbers for each case.

**Funding:** The author(s) received no specific funding for this work.

**Competing interests:** The authors have declared that no competing interests exist.

## Introduction

The adequate treatment of both genital and extragenital infections with *Chlamydia trachomatis* (CT) has been given special attention during the past few years [1]. One of the main reasons for this are several studies reporting prolonged detection of CT after treatment with azithromycin, especially in case of rectal CT [2, 3]. Concerns about potential treatment failure is particularly raised because of the lower efficacy of the single dose azithromycin regimen [2–4], which is still one of the current first-line therapies for the treatment of uncomplicated urogenital CT infections in Europe [5], although in some countries this is being replaced by doxycycline [6, 7].

Efficacy might be compromised due to pharmacokinetic factors [8, 9], but could also be the result of microbial properties such as CT microbial load [9–11], CT specific life cycle dynamics [11] including persistent growth forms (characterized by aberrant reticulate bodies) [11–13], or antimicrobial resistance [9, 14], including transient resistance due to reduced chlamydial viability or due to loss of resistance upon passage [12].

The main mechanisms of macrolide resistance are through modification of the antibiotic target site, through efflux of the antibiotic, by drug inactivation or through prevention of entrance into the bacterial cell by changes in the permeability of the membrane or cell wall [15, 16]. Target modification can be classified in changes in rRNA, either through methylation of naked 23S rRNA or mutation of the 23S rRNA gene, or in ribosomal proteins L4 or L22 [15–17]. To date, there have been no reports of macrolide resistance through efflux pumps, drug inactivation or reduced permeability in CT. However, the function or expression of chlamydial Erm homologues and potential for macrolide efflux of the proteins YgeD and YjjK are unknown [17].

Mutations in the peptidyl transferase region of the 23S rRNA gene were initially reported in 2004 by Misyurina et al. in four *in vitro* macrolide-resistant clinical isolates [14]. These four macrolide resistant strains harbored macrolide resistance associated mutations (RAMs) A2058C and T2611, which are known sites in the peptidyl transferase loop prone to changes causing macrolide resistance in a wide range of bacteria [18]. Additionally, triple mutations were detected in a non-conserved region of the ribosomal L22 proteins in 3 out of 4 of the macrolide resistant strains. However, because these mutations were also discovered in two strains without *in vitro* macrolide resistance, these L22 mutations were not considered responsible for the macrolide resistance [14].

To date, reports linking CT treatment failure to laboratory confirmed resistance/reduced susceptibility are rare [19–24], and all available reports have limited patient numbers in whom azithromycin treatment failure is suspected (n<15). A possible complicating factor is the limited surveillance on chlamydial resistance due to the lack of cell culturing facilities, the absence of a gold standard MIC determination as well as the absence of resistant reference strains for method validation, the relatively low sensitivity of chlamydial culture, the labor intensity and the extended test turnaround time [25]. Furthermore, laboratory procedures have been suggested to heavily impact results [26–28]. Lastly, there might be serovar specific differences [29].

One method to partly overcome these difficulties and to understand the magnitude of the potential problem of CT-azi-resistance, would be to assess genotypic resistance. Interestingly, the recent report by Shao *et al.* suggested that treatment failures could be better explained by the presence of resistance genes than the minimum inhibitory concentration results (MIC) in patients with urogenital CT infections, and emphasized the need for genetic antimicrobial resistance testing [22]. However, previous studies reporting macrolide resistance associated mutations (RAMs) have mostly investigated cultured chlamydial strains [14, 22, 23, 30], and

**Table 1. Designed primers for the *Chlamydia trachomatis* 23S rRNA gene.**

| Primer | Sequence (5'– 3') |
|---|---|
| Ct23S-M1-F | GACTATGGAACGATAGGAGCC |
| Ct23S-M2-F | CATGAATCTGGAAGATGGAC |
| Ct23S-M1-R | CTCTACTCGTGATTGCCAACC |
| Ct23S-M2-R | GTCTACATGGAGTCTCATTGG |

previously used 23S rRNA PCR assays are not eligible for direct use on clinical samples or have not been evaluated directly on clinical samples to the best of our knowledge. In order to facilitate the assessment of genotypic resistance and to partly overcome the limitations of phenotypic susceptibility testing in CT, resistance surveillance would benefit from a genotypic assay which can be used directly on urogenital and rectal samples, without the need of prior culture. A genotypic assay enables high throughput screening of RAMs in different populations, genotypic assays are generally more sensitive than traditional chlamydial culture methods and less technically demanding, and likely are more accessible for laboratories than cell culture facilities. Therefore, our aim was to develop an assay of this kind, to enable the assessment of macrolide RAMs in the chlamydial 23S rRNA gene. And to evaluate this assay in a convenience sample of clinical cases from a previously performed prospective cohort study of which a subset remained CT nucleic acid amplification test (NAAT) positive during a 8 week follow-up period post azithromycin treatment [2].

## Methods

### Ethics statement

Participants provided written consent for future use of samples. The consent procedure was approved by the Medical Ethics Committee at the VU University of Amsterdam (2009/154, CCMO The Hague: NL28609.029.09).

Clinicaltrials.gov Identifier: NCT01448876.

### In-silico analysis

Primers to amplify the region surrounding bases A2058C, C2452A and T2611C (*E coli* numbering) of the 23S rRNA gene of CT were designed (Table 1, Fig 1). 23S gene sequences of CT available in the National Center for Biotechnology Information GenBank database (NCBI GenBank; www.ncbi.nlm.nih.gov/GenBank/) were obtained (2021, January), and compared to frequently occurring vaginal bacterial species, including *Lactobacillus iners*, *Lactobacillus*

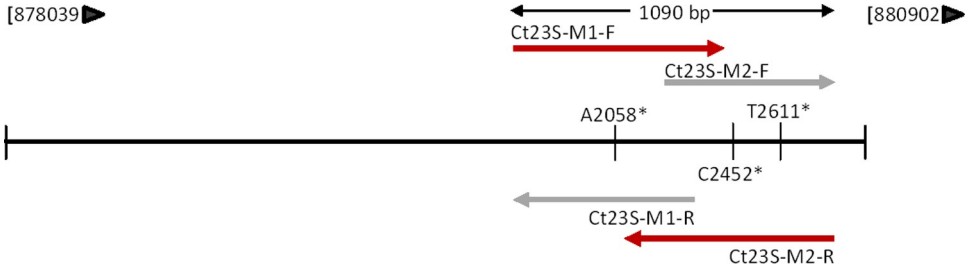

**Fig 1. *Chlamydia trachomatis* 23S rRNA gene with primer sites.** *Chlamydia trachomatis* 23S rRNA gene NC_000117.1 (878039–880902). *: The positions displayed with an asterisk are macrolide resistance associated mutations (*E. coli* numbering).

*crispatus*, *Lactobacillus gasseri*, *Streptococcus agalactiae*, and gut microbes *Bacteroides fragilis*, *Prevotella melaninogenica*, *Enterococcus faecalis*, *Escherichia coli*. Sequences were aligned using Clustal Omega Multiple Sequence Alignment (https://www.ebi.ac.uk/Tools/msa/clustalo/).

To check for the *in silico* specificity of the primers, two approaches were followed. First, the Ct23S-M1-F and Ct23S-M2-R primers were aligned separately against bacterial sequences in NCBI's database using the Basic Local Alignment Search Tool (BLAST, Nucleotide collection (nr/nt), https://blast.ncbi.nlm.nih.gov/Blast.cgi), with exclusion of CT associated sequences. Second, the PCR amplicon was BLASTed (BLAST, Nucleotide collection (nr/nt), NC_000117.1 position 879738–880827). All genetic sequences with ≥95% sequence identity were further analyzed for potential cross-reactivity with the forward and reverse primers, and compared with a CT 23S rRNA gene for differentiating sites (NC_000117.1, 879738–880827). Lastly, internal sequencing primers Ct23S-M2-F and Ct23S-M1-R were designed, to increase the sequencing overlap in the target region of interest (Table 1, Fig 1).

## Analytical specificity and sensitivity

In addition to the *in-silico* analysis, an analytical specificity analysis was performed on 40 CT negative swabs in cobas® PCR medium (Roche Diagnostics, Indianapolis, USA; 20 vaginal and 20 rectal swabs), and a panel of bacterial strains to check for potential cross-reactivity. The panel of bacterial strains were selected based on the potential presence in the urogenital or gastrointestinal tract or genetic similarity with CT (i.e. *Chlamydia* species), and included bacterial cultures, clinical samples or DNA. The bacterial cultures consisted of: *Lactobacillus iners* (1), *Lactobacillus gasseri* (1), *Lactbacillus jensenii* (1), *Streptococcus agalactiae* (1), *Gardnerella vaginalis* (1), *Enterococcus faecalis* (1), *Escherichia coli* (1), *Bacteroides fragilis* (1), *Prevotella denticola* (1), *Clostridioides difficile* (1), *Staphylococcus aureus* (1), *Pseudomonas aeruginosa* (1) and *Chlamydia muridarum* (1; cultured in HeLa cells). The clinical samples included 2 respiratory samples with PCR proven *Chlamydia pneumoniae* (cycle thresholds 31 and 33) and 2 PCR confirmed vaginal swabs with *Mycoplasma genitalium* (cycle thresholds 30 and 34). Lastly, the DNA samples consisted of *Chlamydia suis* DNA (Ct 15), *Ureasplama urealyticum* and *Ureaplasma parvum* DNA (Ct < 25). Analytical sensitivity was analyzed by serial diluting a serovar D CT culture in cobas® medium (dilutions: 10e8 – 10e1 IFU/mL), with each dilution assessed with the 23S primers in trifold.

## Clinical evaluation

Clinical samples were obtained from 52 patients (59 infections) from a previously described prospective cohort study [2]. In short, a convenience sample of patients who visited the STI clinic of South Limburg, The Netherlands, and who were CT NAAT positive at the urogenital or anorectal site before azithromycin treatment, were recruited. Of note, anorectal testing in women only occurred in case of unprotected anal sex in the past 6 months or current anal symptoms. CT screening was performed with the Aptima system (Aptima CT, Gen-Probe, San Diego, USA) using 400μL of sample [2]. Multiple time-sequential measures of 16S rRNA (Aptima CT), DNA and quantitative load were assessed at 18 pre-defined time-point over an 8 week period, as shown in Fig 2. DNA testing was performed with an in-house PCR targeting the cryptic plasmid, using 10μL of prepared sample [2]. The CT load was expressed as inclusion-forming units (IFU) per mL based on defined serial dilutions of CT cultured in HeLa cells (100 IFU to 0.001 IFU) [2].

Samples at baseline (t0) were taken pre-treatment, subsequent samples were post-treatment (t1-t17). All swabs were self-taken. At intake and at the end of weeks 4 and 8, patients completed questionnaires on demographics, sexual behavior, menstruation and symptoms in the

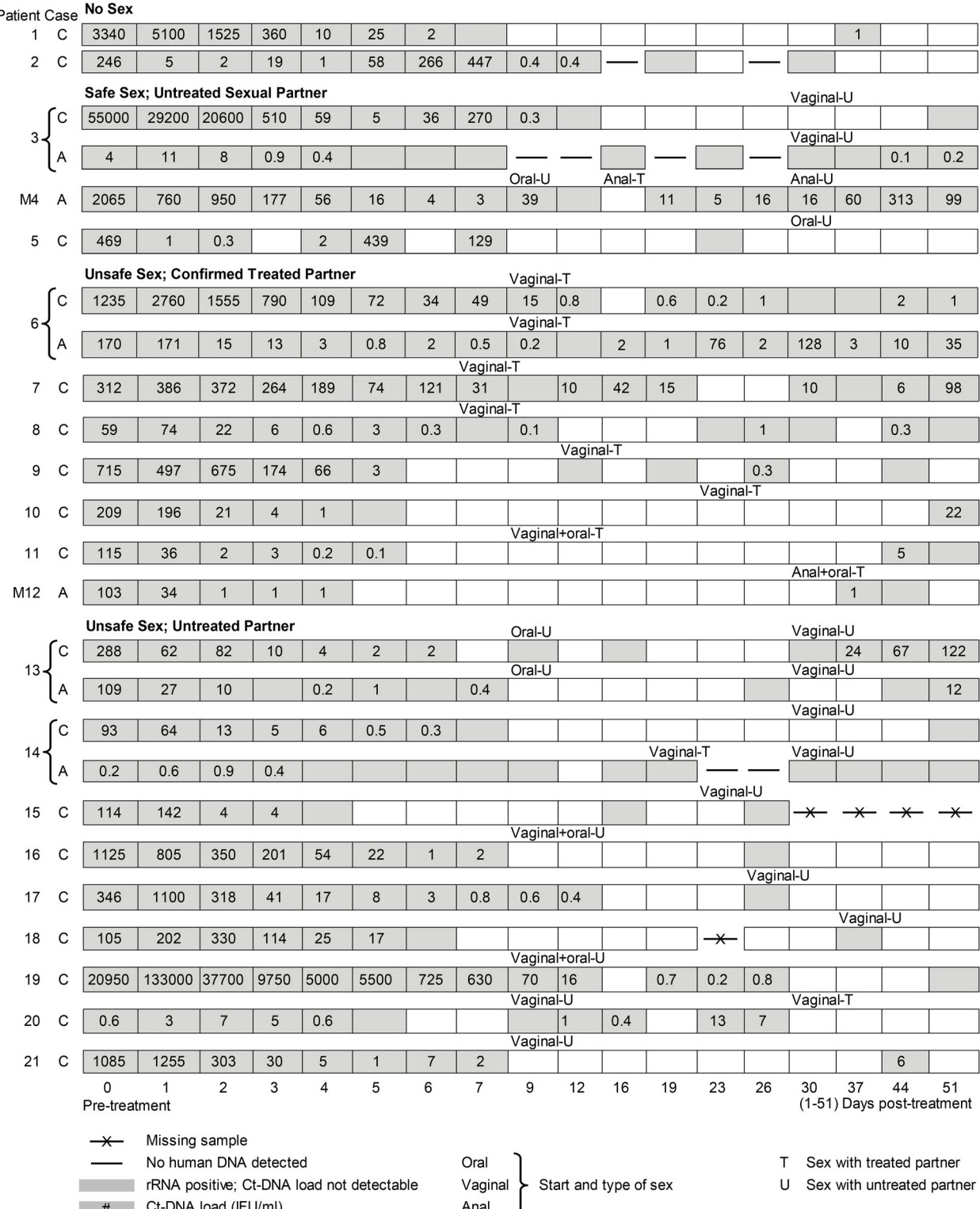

**Fig 2. Multiple time-sequential measures (t = 18) of rRNA, DNA and quantitative load in *Chlamydia trachomatis* cases.** *Chlamydia trachomatis* positivity and load among cases of cervicovaginal (C) and anorectal (A) infections in female and male (M) patients by sexual behavior. This Figure is reused content originally published by Dukers-Muijrers et al [2]. doi:10.1371/journal.pone.0081236.g003.

previous 4 weeks, enabling assessment of CT re-exposure risk [2]. No sex or sex with a tested negative partner only, sex with a treated partner only, or safe sex with a new (considered untreated) partner were considered as low risk of sexual re-exposure. In case DNA isolates could be obtained from 23–51 days post-treatment, serovar determination was performed by OMP1 gene sequencing [2]. Samples from subjects reporting unsafe sex with an untreated partner (Fig 2) were further typed by multilocus sequence typing (MLST) [2]. All t0 samples for which a 23S rRNA sequence could be obtained, were included (i.e. 49 samples, 45 patients; no 23S sequence data were obtained from case 11 and 20, Fig 2). Moreover, 6 follow-up samples were included from 4 patients who demonstrated upward load trends despite treatment and being assigned to the low risk of reinfection group (Fig 2: case 2 t6, case M4A t15 and t16, case 6A t13 and 16, case 7C t16). Of these cases, the latter three remained consistently positive. Sequence typing revealed the same type at intake and follow-up for each case. In combination with the low re-infection risk, these cases were assigned to be possible treatment failures. The sample sequence data were analyzed for the occurrence of RAMs at positions A2057, A2058, A2059, C2452 and T2611 (i.e. *E. coli* numbering; NC_017625, 456981–459884). Other nucleotide variations with a sufficiently high sequence quality (i.e. QV-value >20 in at least 2 reads) were inspected for their occurrence in GenBank and potential relevance regarding macrolide susceptibility. The frequency of RAMs and called nucleotide variations in the clinical samples were compared to gene sequences available from NCBI BLAST's Nucleotide (nr/nt) database, RefSeq Genome Database and Whole-genome shotgun contigs database (January 2021). The GenBank chlamydial 23S sequences were required to have at least 95% sequence identity to the reference genome (23S NC_000117.1, 878039–880902) to control for low quality sequences. Both the obtained GenBank sequences and sequences from the clinical samples were uploaded in a BioNumerics version 7.6 database (Applied Maths NV, bioMérieux, Sint-Martens-Latem, Belgium). The GenBank sequences, available through the accession numbers in the S1 Table, were aligned using the BioNumerics software, after which frequencies of nucleotide variations were counted.

## DNA extraction, PCR amplification and sequencing

DNA was extracted from cultured or clinical samples using Qiagen QIAamp DNA Mini Kit (Qiagen, Hilden, Germany), according to the manufacturers protocol. 200μL of sample was used as input volume, or, 200μL bacterial suspension (0.5 McFarland, 10x diluted in cobas® medium). For every DNA extraction, a positive control and negative control were included and subjected to DNA isolation, i.e. a cultured CT 10e5 IFU/mL and PCR grade water respectively. Amplifications were performed on a Biometra T3000 Thermocycler (Biometra, Westburg, The Netherlands). The PCR was performed with 25μL final PCR volume using 12.5μL AccuStart™ II PCR SuperMix 2x (Quantabio, Beverly, Massachusetts, USA), the CT23S-M1-F and CT23S-M2-R primer (Table 1) at a final concentration of 400nM each, 5.5 μL H2O and 5μL DNA. The PCR started with 5 min denaturation at 95˚C, followed by 40 cycles of 40s at 95˚C, 40s at 60˚C and 90s at 72˚C, and a final extension of 10 min at 72˚C. A 1090 bp fragment of the V-region of the 23S rRNA gene was amplified. PCR products were analyzed on 1% agarose gel. Bands were visualized by staining the gels for 10 min with Ethidium Bromide and visualized with UV transillumination (Uvitec Mini HD9, Alliance UVItec Ltd., Cambridge, UK). Band sizes were compared to a 100bp electrophoresis ladder. PCR products with a single positive band with the approximate 1000bp size were used for sequencing. PCR products were first purified using the STRATEC Universal Nucleic Acid Purification kit (STRATEC Molecular GmbH, Berlin, Germany), according to the manufacturers protocol. Sanger sequencing was performed for each band-positive sample with the 4 primers described

in Table 1. The reactions were carried out with a total volume of 10μL per sample, containing 5.5μL H2O, 1μL BigDye® Terminator v1.1 mix and 1.5μL Sequencing Buffer (Thermo Fisher Scientific, Waltham, Massachusetts, USA), 1μL of 200 nM primer and 1μL purified PCR product. The following PCR profile was used: 1 min denaturation at 96˚C, followed by 22 cycles of 10s at 96˚C, 10s at 55˚C and 3min at 60˚C. Sanger sequencing was performed on an Applied Biosystems® 3730 DNA Analyzer (Thermo Fisher Scientific). Sequence files were uploaded in a BioNumerics version 7.6 database, and sample reads were aligned with each other using the BioNumerics software. Mutations were called only when present in at least 2 high quality reads, defined as a Quality Value (QV) of >20. The 23S sequences were uploaded in NCBI GenBank, and available through accession numbers OM320821 to OM320875 (S3 Table).

## Results

### Technical evaluation

All accessions acquired through BLASTing the Ct23s-M1-F and Ct23s-M2-R primers with ≥17/21 nucleotides overlap were inspected (January, 2021). No bacteria other than *Chlamydia* species had considerable overlap with both the forward and reverse primer. Bacterial sequences with potential overlap with either the forward or reverse primer were compared against the CT 23S rRNA NC_000117.1: With the exception of *Chlamydiifrater* species (96% percent identity), a Chlamydia species established in flamingos [31], neither of these bacterial sequences had a matching overlap above 90% (maximum overlap: *Parachlamydia acanthamoebae* 981/1090 matches and *Waddlia chondrophila* 973/1090 matches compared to NC_000117.1 position 879738–880827). All other sequences acquired through amplicon BLASTing with ≥95% sequence identity originated from *Chlamydia species* (November, 2020). Cross-reactions with of *C. suis* and *C. muridarum* with Ct23S-M1 and Ct23S-M2-R were anticipated. Therefore, discriminating regions within the 23S rRNA sequence for these species were analyzed (NCBI's BLAST nucleotide collection available up to December 2020) and compared to our local database. These discriminating positions comprised of a combination of T1786G, T2110A, A2111G, T2124G, A2132G, A2143C and T2681C for *C. suis* relative to the CT 23S rRNA gene (NC_000117.1, 878039–880902) and C1702T, G1706C, G1715T, T2119C, T2124G, A2132G and C2707A for *C. muridarum* (accession numbers available in the S2 Table). Neither of these discriminating nucleotide variations occurred in our local database. For the remaining *Chlamydia* species, the Ct23S-M1-F primer sites differed between 7 and 11 nucleotides with their respective 23S rRNA target regions. Analytical specificity: The *Chlamydia muridarum* culture and *Chlamydia suis* DNA both caused a false-positive result, as evidenced by gel electrophoresis. Neither of the other bacterial strains in the analytical specificity panel caused a false positive results, nor did the 20 CT negative vaginal and 20 rectal swabs. Regarding the analytical sensitivity, the PCR was able to detect the *Chlamydia trachomatis* serial dilutions 10e8 – 10e3 IFU/mL 3 out of 3 times, and 10e2 IFU/mL 2 out of 3 times. None of the 10e1 IFU/mL concentrations was detected (i.e. 0/3).

### Clinical evaluation

The novel assay was applied to study samples from 45 patients before and 4 patients before and after treatment with azithromycin. The results are shown in Fig 3, showing that novel and known mutations were detected in the 23S rRNA DNA sequences. Macrolide RAMs were not detected in any of the clinical samples, including the follow-up samples from three cases with possible antimicrobial treatment failure. The analysis of RAMs include 23S rRNA positions A2057 and A2059, not shown in Fig 3. Novel mutations were A1817G (QV-values: 52 and 55)

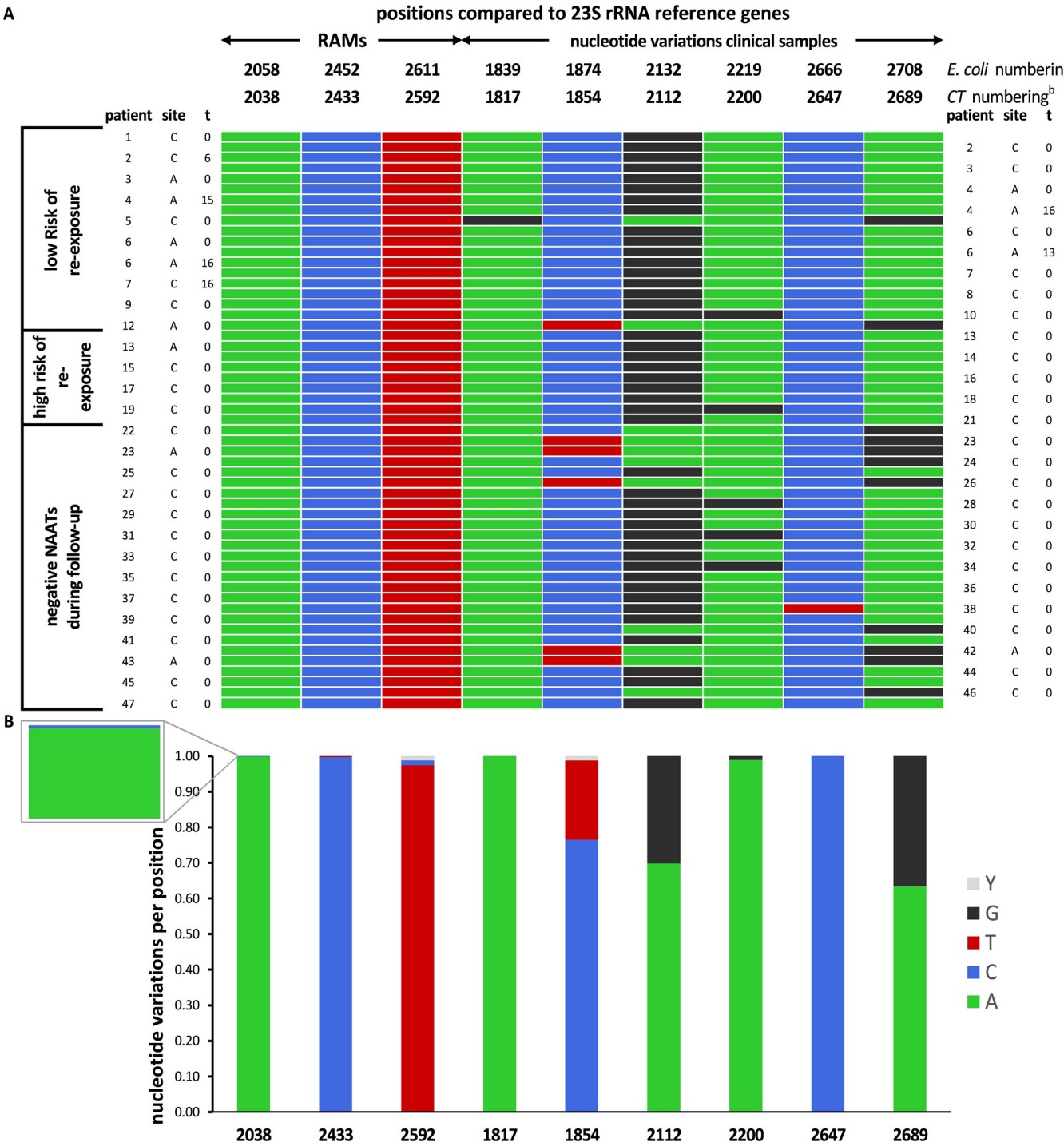

**Fig 3. Resistance associated mutations in the Chlamydia trachomatis 23S rRNA gene.** Resistance associated mutations (RAMs) in the CT 23S rRNA gene and nucleotide variants in a clinical sample of azithromycin treated cases, and frequency of these mutations in NCBI GenBank. **A.** Known macrolide RAMs and nucleotide variations found in the clinical samples, both displayed in *E. coli* numbering ([a] NC_017625, 456981–459884) and relative to CT 23S rRNA gene ([b] NC_000117.1, 878039–880902). A1817G and C2647T are novel mutations. Sequences are available through accession numbers OM320821 to OM320875 (S3 Table). Site of infection: C is a cervicovaginal infection. A is an anorectal infection. **B.** Occurrence of these mutations (both RAMs and nucleotide variants) in NCBI GenBank. Nucleotide frequencies (A/C/G/T/Y) are relative to 23S rRNA gene NC_000117.1 (878039–880902). Sequences were obtained from the NCBI BLAST Nucleotide (nr/nt), RefSeq Genomes and Whole-genome shotgun sequencing databases (January 2021). The zoomed plot focuses on position A2058, critical for macrolide binding in the peptidyl transferase center.

and C2647T (QV-values: 44 and 54), which correspond to A1839G and C2666T in *E. coli* (NC_017625, 456981–459884).

## Discussion

Research in the area of antibiotic resistance in CT has been very limited due to the difficulties in culturing this micro-organism. Genotypic detection of genes or mutations resulting in resistance could facilitate the surveillance of antibiotic resistance in CT. However, previously used 23S rRNA PCR assays have not been evaluated directly on clinical samples or are not eligible for such use, to the best of our knowledge. This study has developed and implemented a *Chlamydia* specific 23S rRNA gene PCR assay for assessment of the presence of known macrolide RAMs in urogenital and rectal samples, without the need for prior culture, facilitating a high throughput and less technically demanding option for susceptibility screening than CT culture. The intended purpose for direct use on clinical samples is supported by the systematic *in silico* analysis and analytical specificity testing which did not demonstrate false-positive results in 40 CT negative swabs (i.e. 20 vaginal swabs, 20 rectal swabs), nor with any of the bacteria used in the specificity panel, with the exception of *Chlamydia suis* and *Chlamydia muridarum*. Applying the assay to clinical samples confirmed that detection was restricted to CT specific sequences, as no *C. suis* and *C. muridarum* specific mutations were detected. Moreover, *C. muridarum*, a rodent pathogen, is unknown to cause disease in humans [32]. For *C. suis* on the other hand, zoonotic transmission has been described among pig farmers, with culturable *C. suis* from conjunctival, nasal, pharyngeal and stool samples [33] and from eye swabs of Belgian pig abattoir workers [34]. From a One Health perspective, *C. suis* might become of future interest as, contrary to CT, stable phenotypic resistance has been described, albeit for tetracyclines [35]. However, in human clinical samples, this cross-reactivity is considered of limited importance.

Using the assay to study 49 samples (45 patients) before onset of azithromycin treatment, of which 21 patients with intermittent or low load detection up to 8 weeks post-treatment (Fig 2) did not detect any 23S rRNA gene mutations known to be associated with resistance. Moreover, three cases with possible antimicrobial treatment failure, who had upward trending DNA loads over time, detectable rRNA, a similar *Chlamydia* sequence type and low risk of reinfection, did not demonstrate any macrolide RAMs (case 4A, 6A, 7C; Fig 2). The inability to detect macrolide RAMs in these cases is at least suggestive of causes other than antimicrobial resistance, especially against the background of the scarcity of reports on antibiotic resistance in CT. However, there are still two factors to take into consideration. First, mutations in the 23S rRNA in chlamydiae have been suggested to impose a competitive disadvantage [30, 36]. As a heterotypic CT might benefit from 23S rRNA gene mutations during antibiotic treatment compared to the wild type population, wild type populations might re-establish once azithromycin has washed out. Second, against a high background of wild type sequences, the current setup of the assay does not allow for detection of low abundance heterotypic resistance. Therefore, the assay can be modified in future studies and can be used in digital droplet PCR or next generation sequencing approaches more suitable for minor variant detection. Moreover, the current results are limited by the small sample numbers. A larger sample size, a more sensitive method and testing additional time points would provide more definite reassurance of absence.

Lastly, two novel mutations in the 23S rRNA were detected: A1817G and C2647T (accession numbers OM320836 and OM320849 respectively), which correspond to positions 1839 and 2666 in *E. coli*. The first has not been previously described in literature, to the best of our knowledge. The latter has been described in *E. coli*, in which C2666U mutations increased the

levels of translational errors [37]. Moreover, both of these mutations occur outside the peptidyl transferase loop of Domain V of the 23S rRNA [14, 16]. It therefore seems less likely that these mutations are of clinical importance.

In conclusion, although reports on macrolide resistance in CT are rare, screening hereof is hampered by the technical difficulties that come with phenotypic resistance testing in CT. In order to facilitate genotypic resistance surveillance directly from urogenital and rectal samples, without the need for prior culture, a 23S rRNA gene PCR assay was developed, and used to evaluate a sample of clinical cases. Although no known macrolide RAMs were detected in the clinical cases, this assay allows future genomic macrolide resistance surveillance in CT, and increases accessibility for other laboratories to partake herein.

## Supporting information

**S1 Table. CT 23S rRNA gene sequences available from NCBI BLAST's Nucleotide (nr/nt) database, RefSeq Genome Database and Whole-genome shotgun contigs database (January 2021).** The GenBank chlamydial 23S sequences were required to have at least 95% sequence identity to the reference genome (23S NC_000117.1, 878039–880902) to control for low quality sequences.
(TXT)

**S2 Table. *Chlamydia suis* and *Chlamydia muridarum* 23S rRNA gene sequences available from NCBI BLAST's Nucleotide (nr/nt) database (December 2020).**
(TXT)

**S3 Table. Cases with corresponding accession numbers (i.e.** OM320821 to OM320875).
(TXT)

## Acknowledgments

We would like to thank Edou Heddema (Zuyderland Medical Center, Sittard-Geleen, The Netherlands) for providing the *Chlamydia suis* DNA.

## Author Contributions

**Conceptualization:** N. H. T. M. Dukers-Muijrers, P. F. G. Wolffs.

**Data curation:** N. H. T. M. Dukers-Muijrers.

**Formal analysis:** J. M. van Niekerk, M. Lucchesi.

**Investigation:** J. M. van Niekerk, P. F. G. Wolffs.

**Methodology:** M. Lucchesi, P. F. G. Wolffs.

**Project administration:** N. H. T. M. Dukers-Muijrers.

**Resources:** C. J. P. A. Hoebe.

**Supervision:** N. H. T. M. Dukers-Muijrers, P. F. G. Wolffs.

**Validation:** M. Lucchesi.

**Visualization:** J. M. van Niekerk.

**Writing – original draft:** J. M. van Niekerk.

**Writing – review & editing:** J. M. van Niekerk, I. H. M. van Loo, S. A. Morré, C. J. P. A. Hoebe, N. H. T. M. Dukers-Muijrers, P. F. G. Wolffs.

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
