## [Decision Letter · Decision Letter 0]

23 Dec 2021

PONE-D-21-37083Direct assessment of mutations in the 23S rRNA gene encoding azithromycin resistance in Chlamydia trachomatisPLOS ONE

Dear Dr. van Niekerk,

Thank you for submitting your manuscript to PLOS ONE. After careful consideration, we feel that it has merit but does not fully meet PLOS ONE’s publication criteria as it currently stands. Therefore, we invite you to submit a revised version of the manuscript that addresses the points raised during the review process.

Please address all concerns of reviewer 1. In addition, please address the following issues:

1) Under Clinical Evaluation in Methods, please briefly describe the methods that were used to determine CT positive clinical samples, CT quantitative load (including how IFU/mL was determined) and Ct typing, even if these methods are in reference 2, as this will facilitate a more complete understanding of the data.

2) Although Figure S1 is reproduced from reference 2, please include as a main figure with appropriate permissions.

3) Please change ‘golden’ to gold; change ‘to best of our knowledge’ to ‘to the best of our knowledge.’ 

Please submit your revised manuscript within one month. If you will need more time than this to complete your revisions, please reply to this message or contact the journal office at plosone@plos.org. Please include the following items when submitting your revised manuscript:

We look forward to receiving your revised manuscript.

Kind regards,

Deborah Dean, M.D., M.P.H.

Academic Editor

PLOS ONE

Journal Requirements:

We will update your Data Availability statement to reflect the information you provide in your cover 

Reviewers' comments:

Reviewer's Responses to Questions

**Comments to the Author**

1. Is the manuscript technically sound, and do the data support the conclusions?

Reviewer #1: Yes

2. Has the statistical analysis been performed appropriately and rigorously? 

Reviewer #1: Yes

3. Have the authors made all data underlying the findings in their manuscript fully available?

Reviewer #1: Yes

4. Is the manuscript presented in an intelligible fashion and written in standard English?

Reviewer #1: Yes

5. Review Comments to the Author

Reviewer #1: Well written paper from a group with a lot of experience in the field. The investigators looked for mutations in the 23S rRNA gene associated with macrolide resistance in PCR positive specimens of C. trachomatis. These included specimens obtained and baseline and sample of clinical cases where the PCR remained positive for up to 51 days post treatment with azithromycin. No resistance associated mutations (RAMs) were discovered. Persistence of C. trachomatis DNA and RNA, but no viable organisms by culture) has been described for 3 weeks or more, which is why the CDC does not recommend retesting before 4 weeks post therapy (2021 CDC STI Treatment Guidelines ref # 6). These results are reassuring. The same RAMs have also been observed in Mycoplasma genitalium, M. pneumoniae and N. gonorrhoeae. SpeeDx has developed PCR assays to detect these mutations in M. genitalium and M. pneumoniae in addition to detecting the organisms.

I would change the title to “Direct assessment of possible mutations in the 23S rRNA gene encoding macrolide resistance in Chlamydia trachomatis”, as they didn't find any mutations. These mutations also confer resistance to other macrolides including erythromycin.

6. PLOS authors have the option to publish the peer review history of their article (what does this mean?). If published, this will include your full peer review and any attached files.

Reviewer #1: **Yes: **Margaret R. Hammerschlag, MD

---

## [Author Response · Author response to Decision Letter 0]

2 Feb 2022

January 21th, 2022

Dear Editors, 

We thank the editors and reviewers for their input and comments on the manuscript and we have edited the manuscript to address their concerns. In the next section, we’ll provide a point-by-point response to the comments.

1. Please address all concerns of reviewer 1.

Reviewer #1: I would change the title to “Direct assessment of possible mutations in the 23S rRNA gene encoding macrolide resistance in Chlamydia trachomatis”, as they didn't find any mutations. These mutations also confer resistance to other macrolides including erythromycin.

We agree with reviewer 1 and have changed the title of our manuscript, as suggested. 

2. Under Clinical Evaluation in Methods, please briefly describe the methods that were used to determine CT positive clinical samples, CT quantitative load (including how IFU/mL was determined) and Ct typing, even if these methods are in reference 2, as this will facilitate a more complete understanding of the data.

We have added the nucleic acid amplification test used to screen for Chlamydia trachomatis infections, we have described the CT quantification as reported by reference 2, and added the methods used to perform sequence typing. 

3. Although Figure S1 is reproduced from reference 2, please include as a main figure with appropriate permissions.

We have included ‘Figure S1’ as a main figure in the manuscript (Fig 2). We have emailed PLOS ONE for permission regarding reuse of this figure. In the reply we received 19.01.2022, it was stated that the content was published under an open access license (CC-BY). The original creators are credited in the Figure’s caption.

4. Please change ‘golden’ to gold; change ‘to best of our knowledge’ to ‘to the best of our knowledge’.

We have edited this in the current manuscript.

We have edited the manuscript according to PLOS ONE’s style requirements, including file naming. 

The reference list is complete and correct. One reference has been added, as described below (point 10). 

 

7. In your Data Availability statement, you have not specified where the minimal data set underlying the results described in your manuscript can be found. PLOS defines a study's minimal data set as the underlying data used to reach the conclusions drawn in the manuscript and any additional data required to replicate the reported study findings in their entirety. All PLOS journals require that the minimal data set be made fully available.

We have provided all accession numbers required to reproduce Fig 2 in the supplementary file called ‘S1_File’.

We also provided the accession numbers of the Chlamydia suis and Chlamydia muridarum 23S gene used to find discriminatory regions. These accession numbers have been added to ‘S2_File’. All 23S sequences generated in the current study have been submitted to GenBank. We have been provided accession numbers for these sequences (OM320821 to OM320875). Our sequences are currently being examined and processed by the GenBank annotation staff to ensure that it is free of errors or problems. 

8. We note that you have included the phrase “data not shown” in your manuscript. Unfortunately, this does not meet our data sharing requirements. PLOS does not permit references to inaccessible data. We require that authors provide all relevant data within the paper, Supporting Information files, or in an acceptable, public repository. Please add a citation to support this phrase or upload the data that corresponds with these findings to a stable repository (such as Figshare or Dryad) and provide and URLs, DOIs, or accession numbers that may be used to access these data. Or, if the data are not a core part of the research being presented in your study, we ask that you remove the phrase that refers to these data.

We have removed the phrase “data not shown” and added the discriminatory regions under the results section (technical evaluation). The Chlamydia suis and Chlamydia muridarum sequences used to evaluate these discriminatory regions, have been added to the supplementary ‘S2_File’. 

9. While revising your submission, please upload your figure files to the Preflight Analysis and Conversion Engine (PACE) digital diagnostic tool.

All 3 main figures have been uploaded to the PACE tool, and LZW compressed.

10. Attachment ‘Riska macrolide R Cpn 2004.pdf’

Although we were uncertain about the aim of the added attachment (‘Riska macrolide R Cpn 2004.pdf’), we were reminded of this very interesting paper with indeed many similarities with our work on CT. We have added this reference in the introduction, 3rd paragraph. 

We look forward to hearing from you regarding our submission. We would be glad to respond to any further questions and comments that you may have.

On behalf of all authors,

Julius van Niekerk

---

## [Editor Report · Decision Letter 1]

28 Feb 2022

Direct assessment of possible mutations in the 23S rRNA gene encoding macrolide resistance in Chlamydia trachomatis

PONE-D-21-37083R1

Dear Dr. van Niekerk,

We’re pleased to inform you that your manuscript has been judged scientifically suitable for publication and will be formally accepted for publication once it meets all outstanding technical requirements.

Kind regards,

Deborah Dean, M.D., M.P.H.

Academic Editor

PLOS ONE
---

## [Editor Report · Acceptance letter]

2 May 2022

PONE-D-21-37083R1 

Direct assessment of possible mutations in the 23S rRNA gene encoding macrolide resistance in *Chlamydia trachomatis*

Dear Dr. van Niekerk:

I'm pleased to inform you that your manuscript has been deemed suitable for publication in PLOS ONE. Congratulations! Your manuscript is now with our production department. 

Kind regards, 

on behalf of

Dr. Deborah Dean 

Academic Editor

PLOS ONE